

# Evaluating differences in density estimation for central Iowa butterflies using two methodologies

Shane Patterson[1], Jonathan Harris[1], Stephen Dinsmore[1] and Karen Kinkead[2]

[1] Department of Natural Resource Ecology and Management, Iowa State University, Ames, IA, United States of America
[2] Iowa Department of Natural Resources, Des Moines, IA, United States of America

## ABSTRACT

The Pollard-Yates transect is a widely used method for sampling butterflies. Data from these traditional transects are analyzed to produce density estimates, which are then used to make inferences about population status or trends. A key assumption of the Pollard-Yates transect is that detection probability is 1.0, or constant but unknown, out to a fixed distance (generally 2.5 m on either side of a transect line). However, species-specific estimates of detection probability would allow for sampling at farther distances, resulting in more detections of individuals. Our objectives were to (1) evaluate butterfly density estimates derived from Pollard-Yates line transects and distance sampling, (2) estimate how detection probabilities for butterflies vary across sampling distances and butterfly wing lengths, and (3) offer advice on future butterfly sampling techniques to estimate population density. We conducted Pollard-Yates transects and distance-sampling transects in central Iowa in 2014. For comparison to densities derived from Pollard-Yates transects, we used Program DISTANCE to model detection probability (p) and estimate density (D) for eight butterfly species representing a range of morphological characteristics. We found that detection probability among species varied beyond 2.5 m, with variation apparent even within 5 m of the line. Such variation correlated with wing size, where species with larger wing size generally had higher detection probabilities. Distance sampling estimated higher densities at the 5-m truncation for five of the eight species tested. At this truncation, detection probability was <0.8 for all species, and ranged from 0.53 to 0.79. With the exception of the little yellow (*Pyrisitia lisa*), species with median wing length <5.0 mm had the lowest detection probabilities. We recommend that researchers integrate distance sampling into butterfly sampling and monitoring, particularly for studies utilizing survey transects >5 m wide and when smaller species are targeted.

Corresponding author
Jonathan Harris, jpharr@iastate.edu

## INTRODUCTION

Ecologists have often struggled with the need to estimate the probability that an organism is detected during a survey, given that it is present. This concept is widely referred to as detection probability (*Burnham & Anderson, 1984*). Early studies tended to ignore it and

assumed that all organisms were detected during surveys (*Mackenzie, Nichols & Sutton, 2005*). Later work shifted towards developing methods to directly estimate detection probability, which include distance sampling (*Eberhardt, 1968*; *Gates, 1968*; *Buckland et al., 2001*; *Buckland et al., 2004*), multiple-covariate distance sampling (*Marques et al., 2007*), mark-recapture (*e.g.*, *Haddad et al., 2008*; *Pellet et al., 2012*), double observers (*Nichols et al., 2000*; *Koneff et al., 2006*), and a synergy of distance-sampling and double-observer methods (*Kissling & Garton, 2006*). If detection probability is estimated to be less than 1.0, that information is used to correct estimates of density and abundance to account for the fact that some fraction of the population is almost always missed during surveys (*Buckland, Laake & Borchers, 2010*). Still, despite increased awareness of imperfect detection, particularly in the field of conservation biology (*Jarzyna & Jetz, 2016*; *Benoit, Jackson & Ridgway, 2018*), estimation of detection probability is widely lacking (*Kellner & Swihart, 2014*; *Kral et al., 2018*).

The importance of addressing concerns about detection probability has clear and important implications for the conservation and monitoring of populations. Conservation plans often address the need to know the size of the population of interest (*Yoccoz, Nichols & Boulinier, 2001*; *McGill, 2006*; *Farr et al., 2022*), and such estimates repeated in time are useful for estimating the trend of the population (*Buckland et al., 2001*). The conditions for surveying a population can change over time for many reasons—differences in habitat, changes in observers, and a host of other factors. As such, trends that rely on changes in relative abundance may be biased because they assume that detection probability remains constant. Additionally, simulations have suggested that accurately identifying population trends becomes difficult when detectability <80% (*Falaschi et al., 2022*), such as for smaller, more cryptic species. Studies that directly estimate detection probability can greatly minimize this source of bias, resulting in more robust inferences about the population, and leading to more informed conservation actions.

Many types of surveys are used to estimate the size of a population, and one of the most common is line transects (*Buckland et al., 2001*). For butterfly surveys, the standard Pollard-Yates line transect has been the most extensively used means of surveying butterflies (*e.g.*, *Brown & Boyce, 2001*; *Collier et al., 2006*; *Nowicki et al., 2008*). This method involves walking at a slow pace (ca. 10 m/min) along a predetermined line and counting only butterflies seen within a prescribed width, often 5.0 m (*i.e.*, 2.5 m to either side of the observer; (*Pollard & Yates, 1993*)). There are several benefits to Pollard-Yates transects including the continuation of long-term studies (*Van Swaay et al., 2008*) and the ability to count many individuals efficiently (*Isaac et al., 2011*). Additionally, abundance estimates derived from Pollard-Yates surveys can be corrected to reduce the bias of double-counting individuals by averaging counts within sites and summing across  sites (*Pollard & Yates, 1993*; *Gross et al., 2007*; *Haddad et al., 2008*). However, an implicit assumption of these surveys is that detection probability of butterflies is 1.0, or that detection probability is constant across the survey period. Violating this assumption means that estimates of density or abundance may not be comparable, and that any changes detected could result from true changes in the population or from changes in survey conditions (*e.g.*, different observers). Data from Pollard-Yates transects are often converted to density estimates and

used to make inferences about populations. The effectiveness of Pollard-Yates transects to correlate with population size has varied (*Collier et al., 2008*). The subsequent reliability of this method is likely dependent on a variety of factors including target species, and continuity of observers and environmental conditions. Fortunately, there are additional survey methodologies that can be used to directly estimate detection probability and yield "corrected" density estimates.

Distance sampling has been shown to be a beneficial sampling method for butterflies (*Powell, Busby & Kindscher, 2007*; *Moranz, 2010*; *Isaac et al., 2011*; *Kral-O'Brien et al., 2020*), given that the assumption of perfect detection can be tested by assigning a distance (or distance bin) to each individual (*Burnham & Anderson, 1984*). Under this sampling method, detection probability is assumed to decrease as distance to the transect line increases (*Burnham & Anderson, 1984*; *Buckland et al., 2001*; *Buckland et al., 2004*), which allows for butterflies to be sampled at greater distances than 2.5 m used in Pollard-Yates transects. Sampling at greater distances allows for more detections of rarer species and consequently, a more complete estimate of community composition. However, distance sampling is rarely used for butterflies, relative to other survey techniques (*Brown & Boyce, 1998*; *Kral et al., 2018*), and imperfect detection as a concept is widely unaccounted for in the majority of invertebrate papers (*Kellner & Swihart, 2014*). Not accounting for imperfect detection can result in underestimates of true abundance (*Burnham & Anderson, 1984*), which may have implications for conservation or management decisions when accurate population estimates are desired. Consequently, researchers have summarized the need to address sampling bias that stems from variation in butterfly detectability (*Dennis et al., 2006*; *Kéry & Plattner, 2007*; *Haddad et al., 2008*; *Nowicki et al., 2008*).

Species-specific variation in detection probability is likely heavily dependent on morphological characteristics (*Iknayan et al., 2014*). We found only one study that accounted for butterfly detection probabilities using species' morphological traits (*Kral-O'Brien et al., 2020*). They used distance sampling to estimate the effective strip width for butterfly detectability as a function of species' size and color, and found that larger, brighter species had higher detection probabilities. Almost all species with a wingspan under 40 mm had an effective strip width of 2.0−4.8 m (*Kral-O'Brien et al., 2020*), suggesting that some small species may be inadequately sampled *via* Pollard-Yates transects, particularly if they are also dull-colored. Conversely, studies that prioritize sampling larger, brighter species, such as monarchs (*Danaus plexippus*), may meet the assumption of perfect detection using Pollard-Yates transects.

In this study, our objectives were to (1) evaluate butterfly density estimates derived from Pollard-Yates line transects and distance sampling, (2) estimate how detection probabilities for butterflies vary across sampling distances and wing length, and (3) offer advice on future butterfly sampling techniques to estimate population density.

Portions of this text were previously published as part of a thesis (*Patterson, 2016*).

## MATERIALS & METHODS

### Study area and site selection

Our study was conducted at four public properties (Harrier Marsh Waterfowl Production Area, Marietta Sand Prairie State Preserve, McCoy Wildlife Management Area [WMA], and Rock Creek Marsh WMA) in central Iowa, each of which was included in the ongoing Iowa Multiple Species Inventory and Monitoring (MSIM) program (*Kinkead, 2006*). Sites were selected to represent a range of habitats appropriate for butterfly species that inhabit typical habitats (*e.g.*, prairies and thickets) and ensure that our surveys would detect sufficient numbers to estimate detection probability for multiple species. Harrier Marsh (170 ha) and McCoy WMA (177 ha) are located in the Des Moines Lobe of the Prairie Pothole Region, while Marietta Sand Prairie (93 ha) and Rock Creek WMA (343 ha) lie nearby in the Southern Iowa Drift Plain (*Prior, 1991*). Although most of the former prairies, marshes, and savannas in these landforms have been converted to row-crop agriculture (*Reeder & Clymer, 2015*), our study sites collectively encompass a mixture of floodplain wetland, prairie-pothole marsh, upland meadow, restored dry-mesic prairie, and shrubby thickets.

### Butterfly surveys

To eliminate observer effects, a single observer conducted all surveys, which took place between 24 July and 24 August 2014 to coincide with prolific flight periods for many species common to this region (*Schlicht, Downey & Nekola, 2007*). Each of the four sites were visited seven times during this period ($n = 28$ surveys) with an average of 3.67 days (SD = 2.44 days) between visits across all sites. Due to logistical constraints all sites were not visited on the same day, but we attempted to space visits evenly across the survey window for all sites. In accordance with MSIM protocol (*Kinkead, 2006*), based on techniques developed by *Shepherd & Debinski (2005)*, surveys were conducted along a single 400-m long, 5-m wide line that had been placed on the centerline of an established 10.4-ha sampling hexagon at each property. For the Pollard-Yates transect, the observer walked at a steady pace (ca. 10 m/min) down the middle of the transect line and recorded number and species for butterflies detected within the 5-m wide transect corridor. Behavior (*i.e.*, flying, nectaring, resting, basking, mineralizing, ovipositing, and courting) at initial detection of each individual was also recorded. Butterfly nomenclature and taxonomic sequence adhered to those of *Opler, Lotts & Naberhaus (2010)*.

On each site visit, the single observer also conducted unlimited-distance line transects in the opposite direction on the same transect line. All aspects of sampling techniques (*e.g.*, pace during survey) matched the methodology of the Pollard-Yates transects, with one difference: for each individual detected, the observer assigned it to one of eight distance bins based on its perpendicular distance from the transect line (bin $1 = 0 - 1$ m, bin $2 = 1 - 1.75$ m, bin $3 = 1.75 - 2.5$ m, bin $4 = 2.5 - 5.0$ m, bin $5 = 5.0 - 10$ m, bin $6 = 10 - 25$ m, bin $7 = 25 - 50$ m, and bin $8 = >50$ m) during the count. These detection-distance categories were arranged in such a way to allow for comparison to the established 5-m sampling width of the Pollard-Yates line transects (S. Shepherd, Iowa Department of Natural Resources, pers. comm., 2014). To allow butterflies sufficient time to settle following sampling disturbance, we waited 10 min before beginning the second

transect and we alternated the survey type (Pollard-Yates or distance sampling) that was conducted first on a given visit. We completed all surveys between 9:00 a.m. and 6:00 p.m. and during warm temperatures ($\geq$ 20 °C), low cloud cover (<70%), calm winds (<16 km/h), and no precipitation. All weather variables were measured immediately before and after the completion of each transect.

## Data analyses

We selected eight butterfly species that had a sufficient number of detections for our analyses (least skipper (*Ancyloxypha numitor*), cabbage white (*Pieris rapae*), clouded sulphur (*Colias philodice*), orange sulphur (*Colias eurytheme*), little yellow (*Pyrisitia lisa*), eastern tailed-blue (*Cupido comyntas*), monarch, and viceroy (*Limenitis archippus* )). Species were chosen to meet the minimum sample size necessary for analysis (*Thomas et al., 2010*), determined by confidence intervals that failed to overlap with zero, and to represent a range of wing lengths that could contribute to detection probability. We used median wing length as a proxy for overall butterfly size, which was correlated with density estimates post-hoc.

Density was estimated differently for the two sampling approaches. For standard Pollard-Yates transects, we summed the site-level mean counts for each species, divided by the total area surveyed (0.8 ha: 0.2 ha per site) to estimate number of individuals per ha (*Pollard & Yates, 1993*; *Gross et al., 2007*; *Haddad et al., 2008*). For unlimited-distance transects, we used Program Distance (v6.2) to estimate detection probability (and associated sampling coefficient of variation (CV)) along with density for each species by site and across sites. For each species, we adjusted the four models endorsed by *Buckland et al. (2001)* ((1) uniform key function with cosine adjustments, (2) half-normal key with cosine adjustments, (3) half-normal key function with Hermite polynomial adjustments, and (4) hazard rate key function with simple polynomial adjustments). These models demonstrate characteristics that meet the distance sampling assumption of monotonically decreasing probability of detection from the line (*Buckland et al., 2001*). We also included models with truncated datasets, where data were limited to observations within 50 m, 5 m, and 2.5 m. Data truncation in distance sampling can be used to remove outliers that increase error when modeling detection functions (*Buckland et al., 2001*). These truncation distances were chosen to represent the Pollard-Yates distance (2.5 m), a reasonable extension of Pollard-Yates methods (5 m), and a realistic maximum distance of detecting butterflies to the species level (50 m). Model fit was evaluated using the chi-square goodness-of-fit test in Program Distance. We used AIC model selection (*Burnham & Anderson, 2002*) to choose the best approximating model for each species. Detection probabilities at the 5-m truncation were correlated post-hoc with species' median wing lengths as reported from *Opler, Lotts & Naberhaus (2010)*.

We provide varying butterfly density estimates derived from Pollard-Yates and distance sampling. We did not statistically compare Pollard-Yates and distance sampling density estimates because we did not have an estimate of precision from Pollard-Yates samples at the no./ha scale and statistical comparisons of modeled coefficients compounds error (*Murrah, 2020*). However, we focus our discussion on species with a Pollard-Yates density

**Table 1 Pollard-Yates survey counts of eight butterfly species in Iowa, 2014.** Average counts of each species were recorded across seven Pollard-Yates surveys for each site. Site-specific averages are presented with standard deviations in parentheses.

|  | Rock Creek Marsh WMA | Marietta Sand Prairie SP | Harrier Marsh WPA | McCoy WMA |
|---|---|---|---|---|
| Least skipper | 28.57 (17.30) | 3.57 (3.31) | 0 | 0.71 (0.76) |
| Eastern tailed-blue | 0 | 10.14 (6.62) | 3.71 (1.80) | 5.29 (2.29) |
| Little yellow | 0 | 16 (8.41) | 0 | 0 |
| Cabbage white | 6.29 (2.81) | 0.29 (0.49) | 0.71 (1.11) | 1.29 (1.38) |
| Orange sulphur | 0 | 1.86 (2.12) | 3.86 (1.57) | 5.29 (2.63) |
| Clouded sulphur | 0 | 1.29 (1.25) | 2.43 (1.90) | 1.86 (1.46) |
| Viceroy | 1.14 (1.21) | 3.43 (2.15) | 5.43 (4.43) | 0 |
| Monarch | 1.71 (2.06) | 1.57 (0.98) | 0.71 (0.76) | 0 |

that fell outside of the 95% confidence interval derived from distance sampling, as those species may be of greater interest when considering differences between the two methods.

## RESULTS

We recorded a sufficient number of detections for least skipper ($n = 350$), cabbage white ($n = 117$), clouded sulphur ($n = 158$), orange sulphur ($n = 291$), little yellow ($n = 414$), eastern tailed-blue ($n = 247$), viceroy ($n = 88$), and monarch ($n = 301$). Least skipper was the most abundant species recorded but this was primarily driven by high abundance at one site: Rock Creek Marsh WMA (Table 1). The two smallest species, least skipper and eastern tailed-blue, were detected in categories extending only out to the 5-10 m and 10–25 m bins respectively, whereas the other six species were recorded in all eight bins. However, in all instances, the two outermost bins accounted for a small proportion (<10%) of the detections, and the median detection distances occurred in the following bins for each species: least skipper (1−1.75 m), cabbage white (1.75−2.5 m), clouded sulphur (2.5−5.0 m), orange sulphur (2.5−5.0 m), little yellow (2.5−5.0 m), eastern tailed-blue (1−1.75 m), viceroy (2.5−5.0 m), and monarch (2.5−5.0 m). For every species, >80% of individuals were identified as flying, basking, or nectaring at time of detection, with resting, mineralizing, courting/mating, and ovipositing butterflies composing the remainder.

Butterfly detectability varied by distance and species' morphology. At the <2.5 m truncation, which is the standard Pollard-Yates transect width, detectability among all species was >0.90, and for several species was estimated at 1.0. However, species-specific detectability began to decrease at distances >2.5 m (Table 2). Consequently, we focus most of our inferences at the 5.0 m truncation because it provides a more interesting comparison with Pollard-Yates densities. At the 5.0-m truncation, detection probabilities ranged from 0.53 for the least skipper to 0.79 for the much larger, more conspicuous monarch (Table 2). At this distance, the top model for all species included a hazard-rate key and simple polynomial adjustments (Table 2). Additionally, a *post-hoc* analysis also revealed a strong positive correlation ($r = 0.91$) between mean wing length and detection probability at the 5-m truncation, suggesting that detection probabilities were greater for larger species

**Table 2  Top models for eight butterfly species in Iowa, 2014.** For each species, we report the model type, Akaike Information Criterion corrected for small sample size (AICc), species-specific detection probability (p), and associated coefficient of variation (CV). Included with model type is the expansion term, with the number of expansion adjustments in parentheses. Estimates are from distance sampling analyses where data were truncated at a distance of 5.0 m. Only the best model is shown for each species and was used for all inferences (see text for details).

| [a]Species | Model, expansion (no. adjustments of orders) | AICc | p | CV (%) |
|---|---|---|---|---|
| Least skipper | Hazard rate key, cosine (1) | 583.26 | 0.53 | 16.75 |
| Eastern tailed-blue | Uniform key, simple polynomial (2) | 393.76 | 0.57 | 11.91 |
| Little yellow | Hazard rate key, cosine (1) | 408.28 | 0.67 | 16.19 |
| Cabbage white | Hazard rate key, cosine (4) | 182.47 | 0.61 | 16.18 |
| Orange sulphur | Hazard rate key, cosine (1) | 334.96 | 0.66 | 13.45 |
| Clouded sulphur | Hazard rate key, Hermite polynomial (4) | 335.13 | 0.66 | 15.69 |
| Viceroy | Uniform key, simple polynomial (2) | 127.34 | 0.76 | 16.71 |
| Monarch | Hazard rate key, cosine (1) | 544.98 | 0.79 | 16.19 |

**Notes.**
[a]Species are ordered by increasing median wing length (mm) (*Opler, Lotts & Naberhaus, 2010*).

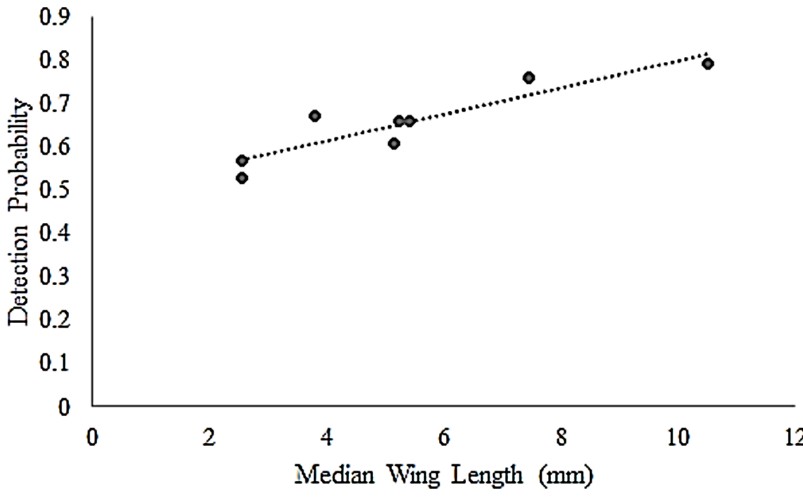

**Figure 1  Correlation of butterfly detectability and wing length.** Species-specific butterfly detection probabilities for eight Iowa butterflies were derived from distance sampling at a 5-m truncation. The resulting detection probabilities correlated ($r = 0.91$) with species' median wing length (mm) (*Opler, Lotts & Naberhaus, 2010*), used as a proxy for size.

(Fig. 1). Detection probabilities at the 50-m truncation were low, ranging from 0.05 (95% CI [0.05–0.06]) for least skipper to 0.14 (95% CI [0.11–0.18]) for monarch.

The corresponding species-specific densities varied by sampling methodology. Densities derived from distance sampling were generally greater than those from the Pollard-Yates transects for multiple species at various truncations. Pollard-Yates densities for clouded sulphur and viceroy were below the 95% confidence intervals of distance sampling estimates at all truncations (Fig. 2). Least skipper, cabbage white, and eastern-tailed blue also had higher density estimates at either the 5-m or 50-m truncation compared to Pollard-Yates

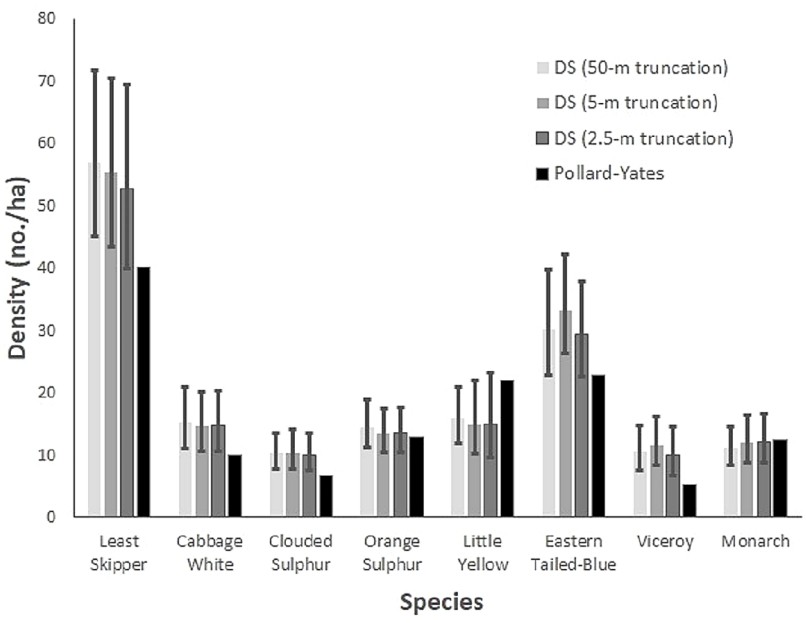

**Figure 2 Butterfly density estimates from two survey techniques.** Species-specific butterfly densities (number/ha) are compared for eight Iowa butterflies. Densities were derived from Pollard-Yates line transects and transects incorporating distance sampling, with data truncated to observations ≤ 2.5 m, 5.0 m, and 50 m. Surveys were performed in Iowa in 2014. Vertical bars depict 95% confidence intervals for distance sampling estimates.

estimates. Generally, species occurring in the highest densities were least skipper, eastern tailed-blue, and little yellow according to distance sampling and Pollard-Yates. However, Pollard-Yates estimates suggest eastern tailed-blue and little yellow occur at similar densities (eastern tailed-blue: 23 no./ha, little yellow: 20 no./ha), while distance sampling predicted more than double the number of eastern tailed-blue compared to little yellow (eastern tailed-blue: 33 no./ha, little yellow: 14 no./ha) at the 5-m truncation (Fig. 2).

## DISCUSSION

Our study found that detection probabilities of eight common, widespread butterfly species in Iowa were at or near 1.0 in the standard Pollard-Yates transect but dropped considerably when the sampling area extended >2.5 m from the line transect. Detection probability was positively correlated with mean wing size and was greatest for the largest, most conspicuous species. Below we discuss our findings in the larger context of methods to estimate butterfly densities, and then comment on how the inclusion of distance sampling can help with conservation and management decisions for this taxon.

### Species-specific detection probabilities

As expected, detectability varied for detections at >2.5 m among the eight species, with an upward trend that generally corresponded to median wing size. This is similar to the detailed findings of *Kral-O'Brien et al. (2020)*, who found that detection probabilities of butterflies in the Great Plains were greater for species with larger wingspans and brighter

colors. The biggest exception in our study was that of little yellow, which produced the third-highest detection probability while having the third-smallest median wing length. Little yellows are generally brightly colored, which is has been shown to increase detection probabilities in butterflies (*Kral-O'Brien et al., 2020*). *Kral-O'Brien et al. (2020)* found that subjective color, defined as the visual apparency against camouflage background, increased the effective strip width. Additionally, little yellows tend to flutter conspicuously just above the tops of *Chamaecrista fasciculata*, a widespread species that serves as the butterfly's favored hostplant in Iowa (*Schlicht, Downey & Nekola, 2007*). This combination of bright coloration and conspicuous behavior may have a compensatory effect on detectability given their relatively small size. Behavioral differences may also be a factor in eastern tailed-blues and least skippers, which are equally short-winged. The eastern tailed-blue exhibits more conspicuous nectaring and breeding behavior, as males patrol and females oviposit high on flower buds of favored host plants like the long-stemmed *Lespedeza capitata* (*Opler, Lotts & Naberhaus, 2010*), which was prevalent along our study transects (Iowa DNR MSIM program, unpublished data). Conversely, the weak-flying least skipper often remains low amongst grasses (*Opler, Lotts & Naberhaus, 2010*). Two very closely related species (orange sulphur and clouded sulphur) produced nearly identical detection probabilities, consistent with their similar size, behaviors, and interrelatedness (*Wheat & Watt, 2008*; *Opler, Lotts & Naberhaus, 2010*; *Dwyer et al., 2015*). Likewise, the brightly colored, wide-ranging viceroy's detection rate was exceeded only by that of its larger, Mullerian co-mimic: the monarch (*Ritland & Brower, 1991*; *Ritland, 1995*).

Despite viceroys having relatively high detection probabilities (at 5-m truncation: $p = 0.76$), the estimated Pollard-Yates density was approximately half the distance sampling estimates. Conversely, monarchs had a similarly high detection probability (at 5-m truncation: $p = 0.79$) as viceroys but monarch density estimates were consistent across all methods. It is unclear why the estimated Pollard-Yates density was lower than distance sampling estimates, given their high detection probability. However, this may simply be due to sampling randomness. In other words, fewer viceroys were observed on Pollard-Yates surveys than on distance sampling surveys only by chance. Similarly, some species with moderate detection probabilities had modeled densities that were similar or smaller than their estimated Pollard-Yates densities. For example, orange sulfur and little yellow detection probabilities were 0.66 and 0.67, respectively, yet distance sampling did not increase the density estimates above the estimated Pollard-Yate densities. Although this is likely another example of sampling randomness, it illustrates the value of having estimates of precision when estimating population densities, such as those provided by distance sampling.

Various studies have demonstrated that the Pollard-Yates line transect is susceptible to sampling bias resulting from differences in detectability (*Dennis et al., 2006*; *Kéry & Plattner, 2007*; *Moranz, 2010*; *Isaac et al., 2011*). Although we did not find variation in detectability close to the line (<2.5 m), we did find considerable differences in interspecific detectability when considering detections at distances exceeding the boundaries of the standard Pollard-Yates transect (>2.5 m). Additionally, given that Pollard-Yates transects necessitate small survey areas, density estimate extrapolation to larger scales is limited

compared to distance sampling, where observations can occur over broader areas and confidence intervals can be derived at any scale. *Moranz (2010)* and *Isaac et al. (2011)* provided the groundwork for utilizing distance data along butterfly transects to estimate population-density. This study adds to the existing literature for species-specific and methodological considerations when sampling bias may be most prevalent. For example, given that smaller species tended to have lower detection probabilities, researchers may seek to extend their sampling area and incorporate detection probabilities into estimates of population density when targeting such species. A useful follow-up to our study might involve assessment of site-specific (*e.g.*, vegetation) effects, as *Brown & Boyce (1998)* and *Haddad et al. (2008)* also documented differences in detectability among sites.

## CONCLUSIONS

In our study, there was minimal variation in detectability at the width (5 m) employed by the traditional Pollard-Yates transect. Consequently, fixed-width transects assuming perfect detection may be adequate for estimating population densities if the target species are large and if transect width is ≤5 m. However, if the research objective is to adequately sample the butterfly community composition or to target smaller, less common species, then incorporating distance sampling with larger transect widths may increase detections and improved density estimates. Narrow transects might not adequately sample large habitat blocks, and avoidance behavior by faster species (*e.g.*, viceroy) might lead to poor and inaccurate counts. By broadening transects and incorporating distance sampling, researchers are likely to detect rarer species and provide more robust estimates of population densities, which may have important implications for conservation actions that rely on accurate population and community estimates. Given the interspecific variation in detectability in our study and other studies (*Moranz, 2010*; *Isaac et al., 2011*), and the observer variation summarized by *Kéry & Plattner (2007)* and *Isaac et al. (2011)*, we recommend incorporating distance sampling whenever possible, especially when transects are >5 m wide and when densities of smaller species (<6.0 mm median wing length) is of interest.

Line transects are straightforward in their implementation and can be used to repeatedly sample butterflies at multiple sites across a broad region in a short window of time. Distance sampling can be easily incorporated into the line-transect framework and analyzed using Program Distance (*Buckland, 2006*), which provides density estimates and associated measures of precision. As such, distance sampling represents an effective tool to survey butterflies and guide the management and conservation of butterfly populations.

## ACKNOWLEDGEMENTS

We thank JA Blanchong, DM Debinski, TM Harms, and KT Murphy for assistance with the study design, analyses in Program Distance, and reviewing this manuscript.

### Funding

This project was funded by Iowa State University and a State Wildlife Grant (T-6-R-5). The funders had no role in study design, data collection and analysis, decision to publish, or preparation of the manuscript.

### Grant Disclosures

The following grant information was disclosed by the authors:
Iowa State University and a State Wildlife Grant:  (T-6-R-5).

### Competing Interests

The authors declare there are no competing interests.

### Author Contributions

- Shane Patterson conceived and designed the experiments, performed the experiments, analyzed the data, prepared figures and/or tables, authored or reviewed drafts of the article, and approved the final draft.
- Jonathan Harris conceived and designed the experiments, analyzed the data, prepared figures and/or tables, authored or reviewed drafts of the article, revised, edited, and formatted the manuscript, and approved the final draft.
- Stephen Dinsmore conceived and designed the experiments, authored or reviewed drafts of the article, and approved the final draft.
- Karen Kinkead conceived and designed the experiments, authored or reviewed drafts of the article, and approved the final draft.

### Field Study Permissions

The following information was supplied relating to field study approvals (i.e., approving body and any reference numbers):

Rock Creek and McCoy are both owned and managed by Iowa DNR. Harrier Marsh WPA is owned by USFWS but managed by Iowa DNR under the Iowa Wetland Management District Agreement (https://www.fws.gov/refuge/iowa-wetland-management-district). Marietta Sand Prairie State Preserve is owned and managed by Marshall County Conservation Board. Permission to survey that property was given to Karen Kinkead by Mike Stegmann (Marshal CCB Director, retired 2022) via email in April 2014 and again in June 2015.

### Data Availability

The raw data are available in the Supplementary Files.

### Supplemental Information

Supplemental information for this article can be found online at http://dx.doi.org/10.7717/peerj.16165#supplemental-information.

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
