# Peer review of "Evaluating differences in density estimation for central Iowa butterflies using two methodologies"

_PeerJ, doi:10.7717/peerj.16165_

## Round 0.1 · original submission · Major Revisions

Dear Dr. Harris,

After this first review round, all three reviewers believe your manuscript has merits to be published in PeerJ. Still, two reviewers raised significant issues that must be improved before the text is published. Therefore, considering all changes that were suggested, I will grant a major suggestion to your manuscript.

Best regards,
Daniel Silva

Reviewer 1 ·

Basic reporting

1. The Introduction needs some re-structuring. Topics are brought up in the beginning (e.g., different types of corrections) that would be better suited later in the Introduction (e.g., L88), and paragraphs seem to cover a lot of issues without a clear take-home message/justification for research.
2. L66 is another area where the topic of the paragraph really seems to change (this happens throughout the manuscript). Really pay attention to what the message is in each paragraph. Then go back and make sure each sentence relates to that message. Here, you start by talking about the need for exact numbers but switch to talking specifically about line transects.
3. Since the Introduction is so short, the authors might want to reconsider what’s important to bring up. Conversely, I could see a way to expand the Introduction by bringing up the additional side of surveying—that not everyone needs to have perfect numbers. If you address these methods to correct, I think you could also address the fact that some surveys don't care if detection/abundance is perfect because they only want an index. Since you are talking about butterflies, the authors should potentially discuss how long-term butterfly surveys across Europe don't use detection probabilities because they have the long-term data. This is a great way to compare how both methods are valid but are better suited to different situations based on survey history.
4. Using dates in the Introduction to talk about methods (L76 and L89) is not very effective for the authors. In the first instance, Pollard walks have been around since at least the 70s and used widely in Europe, so this doesn’t seem entirely true. Additionally, distance sampling still isn’t used widely for butterflies—it just started to be used for butterflies in the 2000s (Kral et al. 2018).
5. L96: I would say the Pollard method hinges on the imaginary box that moves with the observer (5x5x5m). For at least the Moranz and Kral-O'Brien paper, these researchers used line transect methods and adapted them to butterflies, meaning there was no imaginary box and butterflies could be observed at any distance to the side and in front of the observers.
6. L105-106: Save why wing length was selected for the Methods.
7. L166: The authors need to tell the audience what the minimum number of detections are in their study.
8. L205: It seems strange to bring up the models used for each species in Program Distance in the middle of this paragraph—suggest moving it to a more generalized section.
9. L252: It's worth mentioning in the Introduction how Pollard data is typically corrected (long-term use, different equations for calculating corrections based on survey history). A huge thing is that the US just doesn’t have the long-term data that they have in Europe.
10. L255: I would not agree that Moranz and Isaac helped to transform Pollard data into density data-maybe a different word here is more suitable since it’s a bit strong right now. That’s not actually what’s happening. They used a different method (line-transect distance sampling) to calculate density estimates. It’s very similar to Pollard transects, but they aren’t taking numbers from Pollard transects and turning them into corrected density estimates. These authors aren’t the first to use distance sampling for butterflies either if that’s what was meant by the statement.
11. They discuss how the detection probabilities change based on the cut-off (5 m to 2.5 m), but they should include the detection probabilities when possible at each truncation (50 m, 5 m, and 2.5 m). The strength of your data comes from comparing different truncation distances and how that relates to detection probabilities. It would also make it easier to visualize how distance from the observer affects the species observed and the number of detections.
12. Table 1: I don’t find this table particularly helpful, as is. The model results could definitely be a table, but it would make more sense as a supplementary table. I would rather see the detection probabilities of each species at each of the three truncations. The Figure could use images of each of the species with their wingspan listed.
13. Figure 1: Do the authors think the differences between distance sampling and Pollard transects would get larger if they used the 50m truncation? The power of adding a distance measurement is to capture additional data and improve the density estimate. Although you get some of that within 5 m, it seems that it would be even greater with an unrestricted “belt” around the transect. This would explain why monarchs don’t change much between distance and Pollard. Most monarchs aren’t going to be observed that close to someone walking in the field unless they are very careful about looking ahead of the line—that’s why it’s useful to not be restricted. Additionally, what number is being shown in the figure? The average among sites? At a site over the seven visits? More details are necessary. If you use an average, a boxplot may be more informative compared to a bar graph.

Experimental design

1. L127-130: I’m concerned about the number of surveys that were done within a small timeframe. Seven visits within a month could potentially violate one of the assumptions of distance sampling that individuals aren’t double counted (Buckland 2001). While I typically think about this within the same survey, the same individuals could easily be detected if a visit was conducted once every 3-4 days. Did the authors make sure to include some type of repeated measures in their modeling? I forgot how this works exactly in Program Distance, but I know this is often done for bird surveys since they are conducted with the same individuals within a breeding season. Depending on how the visits were dealt with in the model, I would consider only using surveys at the same site that were spaced farther apart in time (e.g., at least 3 weeks).
2. L146: In distance sampling, you can look ahead of the line to make sure you detect them at their initial location. Did the observer do that as well? Or just look within the box in front of them and just extend it out on the sides?
3. L173: I’m having a hard time understanding why the authors don’t think they can run produce errors/statistics on their data. They were able to produce errors in Program Distance, I believe through a bootstrap. Could a similar procedure be used to provide an error for the Pollard data? Additionally, if they are limited by the number of surveys, they should consider using more data if available, so statistics can be conducted (I assume the surveys were completed in more years besides 2014). Otherwise, I don’t find much value in comparing densities between the methods.
4. L185: I would recommend some type of statistical analyses to compare the density estimates produced by the methods. It seems like a great opportunity to compare since the different methods were used at the same site along the same path. Kadlec et al. (2012) [Timed surveys and transect walks as comparable methods for monitoring butterflies in small plots. J Insect Conser] might be a helpful resource to start if you need ideas.
5. L207: The procedure for the post-hoc analysis should be included in the Methods.

Validity of the findings

1. L285: Odd to bring mark-recapture up at the very end. Mark-recapture is another method to create a correction for imperfect detection so it wouldn’t be used in conjunction with distance sampling. Please remove.
2. Since the authors do not use a statistical framework to compare their data, the findings do not have much validity when comparing the density estimates. The findings on detection probabilities for certain species would be valid.

Additional comments

1. I believe they are discussing an important topic, but they do not successfully convey their take-home points or analyze their data in an appropriate way for publication. I strongly suggest going through each paragraph and ensuring the information in each paragraph goes together and helps the story. Additionally, the authors can highlight the necessity of distance sampling by simply showing more of the comparisons in figures and using a statistical framework to compare between cut-off points and methods.
2. L35: Please use scientific name when discussing a specific species.

·

Basic reporting

This is a great study that aims to compare two sample methods for butterfly surveys, and the results may importantly add to the current literature. However, some points raised concerns about the validity of the results and I think they need to be reviewed or at least better described in an updated version, they are:

- Despite the study aims to compare two sampling methods, an important variable (density) was calculated differently in each Pollard-Yates and distance line transect. My recommendation is, if possible, to use the same calculation to estimate the density. If it is not possible, the authors need to add more comments about the equivalence of these different density estimations.

- There was reported a strong association between body size and detectability, and for detection and distance, indicating that small-bodied species and those distantly located from the central line are less detected compared to large-bodied and those located closer to the central line. Because of these factors, I would encourage the authors to provide additional recommendations/comments about the preferred distance (or limiting width) to consider in the distance-line sampling.

I have additional comments regarding the structure of this article and some minor comments as well.

Experimental design

The design is great, my only concern is about the differential calculation of density estimation in each sample method. It needs to be corrected or, at least, better described.

Validity of the findings

I will recommend the authors revise the calculation method of each density estimation just to make sure they are equivalent, or at least comparable.

Additional comments

Some minor comments.

The authors stated (in lines 160-164) they selected eight species based on the number of detection of each species. This information was repeated in the Result section (lines 188-190), because of that I recommend the authors both: complement the information located among lines 160-164, and eliminate the equivalent, located in the Results section (lines 188-190).

The author reports a positive association between body size and detection probability, I recommend the inclusion of a scatterplot (y~x chart) illustrating this association.

·

Basic reporting

Although I cannot judge the grammar, the article was well-written and technically correct.
The introduction and methodology text should be revised to accommodate explanations of how butterfly size may affect detection (more details in the attached PDF).

Experimental design

The question raised in the study is simple and straightforward, but some elements are missing in the methodology (regarding the size of the species) that should be improved.

Validity of the findings

The results are interesting, but I believe the authors could benefit from discussing the similarities or differences between the methods. This way, it is more straightforward to understand when one methodology may be more appropriate, depending on the study organism.

---

## Round 0.2 · Major Revisions

Dear Dr. Harris,

After this second round of reviews, two reviewers still raised a significant amount of issues related to your manuscript that hinder its acceptance for publication at this time. I must say that I agree with most of the issues raised by both of the reviewers. Therefore, I cannot bring you a positive message at this time. Therefore, I will ask you to please take special care regarding the issues raised by reviewers #1 and #3 in the next version of your study.

By the time you are about to resubmit, please do not forget to prepare a rebuttal letter informing the reviewers of all the changes that were applied to the new version of your manuscript and all of those that were not. Please do not hesitate to write in case you have any doubt.

Sincerely,
Daniel Silva

Reviewer 1 ·

Basic reporting

Be careful with the language concerning the various distances used throughout so things are really clear. The measured distance, distance bins, typical truncation used with Distance to create curves (i.e., some data points are removed before creating detection probabilities), and additional truncation to mimic other methods (i.e., only using data points up to a certain distance) needs to be really clear.

Experimental design

Can the authors justify using the confidence intervals from the Distance density to say whether or not the density differs from the Pollard-Yates density? A citation where a study did something similar would be helpful.

The authors have the ability to create an average and variance for the Pollard-Yates densities, even if they don't statistically compare the two densities. Even if there's only 1 transect per site, there are still 4 sites that could be used to create an average for each species, unless the authors don’t think visits were independent. One way to do this would be to average to the site level, since these are visited 7 different times. The corrected densities in Distance could also be stratified to the site level if you wanted to maintain site identity.

Validity of the findings

The objectives are not supported well by the results.
Objective 1: compare butterfly density estimates from Pollard-Yates line transects to distance sampling
To compare the densities, it would be best to initially compare data gathered at the same level and then with the unlimited-width corrected density. So, the Pollard-Yates density compared to the corrected density at 2.5 (creating 5 m total). The figure caption makes this slightly confusing because the authors say truncated at 5 m, which could mean 2.5 m on either side or 5 m on either side (10 m total). Additionally, the authors talk about comparing Pollard-Yates densities to distance sampling within the 2.5-5 m bin only? (L202-203). This needs clarification. To me, the authors should make the same comparison first (2.5 PY to 2.5 Distance sampling). Then, show how both of those compare to the unlimited-width. Looking specifically at the 2.5-5m bin doesn't translate well to any type of method application because an observer wouldn't only look from 2.5-5 m from the line.

Objective 2: estimate how detection probabilities for butterflies vary across sampling distances and wing length.
This would be better accomplished by showing the detection curves created in Distance for each of the eight species. This way, the authors could visually show how steep or shallow curves are based on species size and distance. The table and text used to describe this now is not effective.

Additional comments

The Introduction has been improved during revisions.

·

Basic reporting

I think the authors have made substantial changes which resulted in a higher-quality manuscript.

I would like to clarify my comment regarding the repeated information found in the Results section. In the Methodology section, the authors mentioned that they "selected a set of eight species for analyses." In the Results section, the authors start this section by stating "Eight species had a sufficient number of detections for our analyses". I still think this phrase is not suited for the Results section, once it suggests some criteria were used to select those Eight species. I think it would be more effective if the authors began the paragraph by providing the abundance of each species, instead of reiterating the information about the number of species selected for analyses

Experimental design

I think the information provided by the authors increase the quality of this manuscript

Validity of the findings

no comments

Additional comments

no comments

·

Basic reporting

Some points were unclear in the first review, such as how density was calculated for the Pollard-Yates transects. In this round, the authors performed a major revision of the manuscript, generally agreeing with the reviewers' suggestions. Such modifications improved both the main message of the paper as well as explanations of the methods/analyses used.
To me, the main weakness of this study is the way the authors calculate the density for the Pollard-Yates transect data. Sampling occurred over a period of one month, with an average of three days between visits to the same site. If the observer did not mark individuals [This issue was already raised by reviewer 1], how can one be sure that the same individual was not counted at least twice? Furthermore, the counts were summed to obtain a density index, which can be highly biased by false positives, leading to overestimates. The authors cited in the introduction that are ways to “correct” density estimates (lines 172 – 174), but they did not apply this correction in the study.

Experimental design

no comment

Validity of the findings

The results presented in the study are not sufficient to establish a method comparison study. A method comparison effectively requires a statistical comparison between the methods used, the power of the statistical test, and simulations, as was done in the Haddad et al. (2008) study (DOI: https://doi.org/10.1111/j.1523-1739.2008.00932.x.). Therefore, I suggest changing the title to something more appropriate for the content of the article, such as "Evaluating differences in density estimation for Central Iowa Butterflies using two methodologies". This way, the authors can discuss the results found but not try to indicate which is the best method for sampling butterflies, but within the local context. However, if the purpose of the paper is to compare methods, then the analyses should be redone.

Additional comments

Minor comments

Results
Lines 370 – 372: How did you calculate the probability of detection for the data derived from the Pollard-Yates transects?

Discussion
Lines 486 – 489: I disagree with this statement, and it contradicts another sentence you put in the conclusion of the study [lines 522 - 523]. It is not straightforward to think that organisms with equal detection should result in populations with equal densities in both methods. There are several other processes than stochasticity, such as niche partitioning, and dispersal capacity, that could cause this lower estimate for the Pollard-Yates method.

Lines 505 – 507: What results support the greater importance of including distance sampling when the target is small or low-density butterflies? The Viceroy is a large butterfly with high detection but still showed differences between the estimates from the two models. Also, I would venture to say that density estimates from distance sampling will always be better than the value observed by the Pollard-Yates method because they not only include imperfect detection but also a measure of uncertainty, which is critical for decision-making.

---

## Round 0.3 · Minor Revisions

Dear Dr. Harris,

Congratulations on the hard work you and your co-authors dedicated to improving the manuscript originally submitted. Both reviewers (and I) believe your manuscript has improved a lot. Therefore, please consider your manuscript accepted (in practice) in PeerJ. Nonetheless, you will see that reviewer #1 raised some minor issues that need to be corrected/added to your text before we can formally accept your contribution.

Therefore, please provide the few necessary changes, and as soon as reviewer#1 believes the changes were made to the new version of your text, the MS will be formally accepted for publication.

Sincerely,
Daniel Silva

Reviewer 1 ·

Basic reporting

No comment

Experimental design

No comment

Validity of the findings

No comment

Additional comments

Just a few things to note:

L210 (tracked changes word document version): I really liked the addition of this explanation. This will help bring clarity for the audience.

For Figures 1&2 make sure you put the right truncations in the figure captions (this changes depending on if you look at the figure captions next to the figure vs right before the figure). Figure 1 needs the 5-m truncation added, and Figure 2 needs the 5-m truncation part removed.

L336: I apologize for not bringing this up in an earlier review, but I think it’s important to provide a bit more context here. Make it clearer to the audience that even though the detection probabilities were close to 1 within the PY transects, detection probabilities don’t remain stagnant (what you show with your data and bring up here). What you need to hit home a bit harder is how extrapolating density estimates from PY transects are inaccurate compared to distance sampling. You obviously elude to this, but I just want that to be really, really clear in this section. Just tweaking a sentence or adding one would be perfectly fine. Alternatively, you could illustrate the point with an example--use the density estimates for a species from PY and the 50-m truncation and extrapolate the numbers to some generalized area (e.g., take the largest area surveyed [~350 ha] to show how abundance estimates for the least skipper could vary between 14,000 and >19,000 when using different methods). If you'd go the example route, you could show this earlier in the Discussion when going through some of the species specifics.

L353: instead of "...widths may increase detections.", I suggest "...widths may increase detections and improve density estimates."

L355: Instead of "...might lead to poor counts.", I suggest "...might lead to poor and inaccurate counts.".

·

Basic reporting

no comment

Experimental design

no comment

Validity of the findings

By changing the focus of the work, the results found were consistent with the purpose of the study.

Additional comments

I would like to start by congratulating the authors for the reviews, always trying to answer/question the reviews with a lot of professionalism, and apologizing for the mess with the lines in my minor comments. Nevertheless, the authors managed to correctly clarify the passages I referred to.
I liked the figures, I think it is important to show the variations for the different bins used (Fig. 2). I also liked the table with the values per site, as we can see that "Little yellow" was observed in only one of the sites, which could explain its lower density if imperfect detection is included, or even that "Cabbage white" was the only species detected in all the sites, so I suggest including the table as a material of the main text, as it also helps with the suggestion you leave in the discussion about the effect of sites on detection and consequently on density estimates.

With best regards,
Aline

---

## Round 0.4 · accepted · Accept

Dear Dr. Harris,

I am pleased to inform you that your manuscript has been formally accepted for publication in PeerJ! Congratulations on the hard work you and your co-authors made to reach this decision!

Sincerely,
Daniel SIlva, PhD